# Near-Death Quality of Life in Cancer Patients on Home Parenteral Nutrition

**DOI:** 10.3390/nu17020271

**Published:** 2025-01-13

**Authors:** Paolo Cotogni, Luca De Carli

**Affiliations:** 1Pain Management and Palliative Care, Department of Anesthesia, Intensive Care and Emergency, Molinette Hospital, University of Turin, 10126 Turin, Italy; 2Clinical Nutrition Unit, ASL Città di Torino, 10128 Turin, Italy; luca.decarli@aslcittaditorino.it

**Keywords:** quality of life, home parenteral nutrition, oncology, nutritional support, palliative care, home care, near-death

## Abstract

**Background:** The impact of home parenteral nutrition (HPN) on the quality of life (QoL) of cancer patients has been previously investigated. However, scarce data are available regarding near-death QoL in patients with cancer receiving HPN. This study aims to investigate the changes of QoL in these patients in the last two months before death. **Methods:** This is a secondary analysis of a previous, prospective, longitudinal, observational study. QoL was assessed using the EORTC QLQ-C30 questionnaire. **Results:** Eighty-four adult cancer patients who died on HPN and had filled out the questionnaire between 31 and 60 days (M2) and within 30 days prior (M1) to death were included in this analysis. The questionnaires filled out at M2 and M1 were compared with those filled out by the same patients at HPN start (T0). At M2, there was a significant improvement in both the global QoL and symptoms scales (*p* < 0.001 and *p* < 0.033, respectively), while at M1, a significant improvement in the global QoL scale persisted (*p* < 0.035) compared with T0. **Conclusions:** Our study first reports that HPN, if started early and according to European guidelines, is associated with an improvement in the QoL of patients with cancer even in the last two months before death.

## 1. Introduction

Malnutrition affects up to 80% of patients with cancer, with significant differences related to age, stage, and tumor type [1,2,3]. It is characterized by weight loss with skeletal muscle mass loss, anorexia or reduced oral nutrition, and functional and psychosocial impairment [4,5,6]. In patients with advanced cancer, nutritional status deficits can compromise performance status (PS), tolerance to anticancer treatments, and overall survival [1]. Malnutrition has also been shown to be a significant predictor of worsening quality of life (QoL) in these patients [7,8,9,10]. These findings support the recommendation that medical nutrition therapy should be an integral part of the multidisciplinary and multidimensional approach at every phase of disease care [11]. Specifically, nutritional support must be early and patient-oriented [12].

Home parenteral nutrition (HPN) is an important part of medical nutrition therapy in malnourished advanced cancer patients. According to the current European guidelines, HPN is indicated in patients receiving anticancer treatments if oral nutrition is inadequate, despite counseling and oral nutritional supplements (ONS), and enteral nutrition (EN) is not sufficient or possible [13,14]. In patients in palliative care, guidelines recommend nutritional interventions only after considering together with the patient the prognosis of the disease and both the expected benefits in terms of QoL and potential survival, as well as the burden associated with medical nutrition therapy [15,16].

The impact of HPN on the QoL of patients with cancer has been researched for more than 30 years. Several studies suggest that HPN can positively influence QoL in the first few months. However, scarce data are available regarding near-death QoL in patients with cancer on HPN. The rationale for this study is to fill the gap in the existing literature regarding HPN-associated QoL near death. In particular, the purpose is to investigate the changes in QoL in these patients in the last two months before death.

## 2. Materials and Methods

### 2.1. Study Design

This is a secondary analysis of a previous prospective, longitudinal, observational study carried out in two tertiary referral care university hospitals with a total of more than 1700 beds over a two-year period [17]. The study was approved by our Institutional Ethics Committee. All cancer patients provided written informed consent.

The criteria for including patients in our HPN program followed the recommendations of European guidelines [18,19] and have been previously described [17] (Appendix A).

### 2.2. Procedures

All patients were evaluated at baseline in our hospital by the dietitian and physician in charge of the HPN. Recorded data included weight loss in the past three months, BMI, tumor stage and site, anticancer treatments, and evaluation of the oral residual food intake. PS was classified using the Karnofsky performance status (KPS). Nutritional status was assessed using the scored patient-generated subjective global assessment (PG-SGA) [20].

After initiation of HPN, all patients were carefully followed up by the physician in charge of HPN by structured and regularly scheduled telephone interviews (at least every 15 days) and home visits by the nursing staff and primary care physician (initially every day for 2–3 weeks and then at least every 7 days). After appropriate education, home caregivers delivered the HPN; 24-h telephone support was available for patients, their caregivers, and healthcare providers.

HPN was administered 10- to 14-h-per-day at nighttime through a central venous catheter, using “all-in-one” bags and considering a need of 20–25 kcal/kg per day for bedridden or 25–30 kcal/kg per day for ambulatory patients and 1–1.5 g/kg per day amino acids.

An ambulatory re-evaluation by the dietitian and physician (also including a 24-h oral food recall) was conducted every 30 days after the start of HPN (±5 days). All patients were monitored until withdrawal of HPN or death. HPN was withdrawn when adequate oral nutrition was restored.

### 2.3. QoL Analysis

At baseline and each monthly follow-up visit, QoL was assessed by using the European Organization for Research and Treatment of Cancer Quality of Life Questionnaire Core 30, (EORTC QLQ-C30) Version 3.0, a tool specifically designed to measure QoL in patients with cancer [21]. Specifically, the validated Italian language version was adopted [22]. The EORTC QLQ-C30 is a 30-item questionnaire consisting of (I) a global scale (global QoL); (II) a financial well-being scale (financial impact); (III) 5 functioning scales (cognitive, physical, emotional, role, and social); and (IV) 8 symptom scales (pain, loss of appetite, dyspnea, nausea/vomiting, fatigue, sleep disturbance, diarrhea, and constipation). In our study, the symptoms were aggregated into a global scale called a symptom scale (SYMP). Appetite loss, nausea/vomiting, fatigue, and financial impact were also analyzed individually.

The questionnaire was completed by the patients at the beginning of the HPN, in the presence of a dietitian or a physician if assistance was needed. Thereafter, it was completed by the patients on their own. The raw scores were linearly converted to obtain standard scores in the range of 0 to 100 for each of the functioning and symptom scales. Lower scores in the symptom scales denote better QoL, while higher scores in the functioning and global scales denote better QoL [23].

### 2.4. Statistical Analysis

For this secondary analysis, adult cancer patients who died on HPN and had filled out the questionnaire between 31 and 60 days (M2) and within 30 days prior (M1) to death were included in this analysis. In particular, the questionnaires were divided according to the proximity of the date of filling out to the date of death; those filled out in the 30 days before death were collected in the M1 group, and those filled out between 31 and 60 days before death in the M2 group Subsequently, these questionnaires were compared with those completed by the same patients at the beginning of HPN (i.e., T0) to assess near-death QoL (Figure 1).

All results of continuous variables are reported as the median plus range. Questionnaire scores were nonparametric ordinal data with continuous distribution, and a comparison of scores at different times (M2 vs. T0 and M1 vs. T0) was conducted using Mann–Whitney’s U-Test. All *p*-values were derived by the two-sided exact method at the conventional significance level of 5%. Data were examined in February 2024 by using R V.3.5.3 software (R Foundation for Statistical Computing, Vienna, Austria).

## 3. Results

### 3.1. Patients

One hundred and eleven adult patients with cancer were consecutively enrolled in the primary study and followed till death. No patients were lost to follow-up. Questionnaires from 84 patients who died on HPN were analyzed for this secondary analysis, while questionnaires from 27 patients (24%) whose HPN was withdrawn because appropriate oral food intake had been restored were not included in this analysis.

Patients’ characteristics are depicted in Table 1. All cancer patients had an advanced disease; most had a gastrointestinal tumor; 79% received anticancer treatments; and 83% were cared for by home palliative care services. The major indication for HPN was intestinal (sub)obstruction (79%), and two-thirds were severely malnourished.

All patients had oral residual nutrition (a median of 600 kcal); thus, supplemental HPN supplied a daily median amount of 1000–1250 kcal. The patient-tailored HPN program included 4 to 7 weekly administrations by overnight infusion. The median duration of HPN was 137 days. No deaths could be related to HPN; no major electrolyte disorders occurred. The incidence of catheter-related bloodstream infections (CRBSI) was 0.33 per 1000 catheter-days, mechanical complications were uncommon (0.80/1000), as was venous access devices-related thrombosis (0.08/1000).

### 3.2. QoL Analysis

Patients’ scores for the EORTC QLQ-C30 scales at different time-points with the comparison between T0 and M2 and M1 are reported in Table 2.

In the 60 days before death, there was a significant improvement in both the global QoL and symptom scale compared with the start of HPN (*p* < 0.001 and *p* < 0.033, respectively). In the 30 days before death, a significant improvement in the global QoL scale persisted (*p* < 0.035) compared with T0.

## 4. Discussion

This study first reports that HPN is associated with the maintenance of significantly improved QoL compared with HPN start even near death in cancer patients. Specifically, HPN is associated with improved overall perceived QoL according to the EORTC QLQ-C30 scores in the last two months of patients’ lives.

The present study is partially based on some data collected in our previous study. However, the purpose of our previous study was to evaluate QoL in patients with advanced cancer undergoing HPN and to examine if combination with anticancer treatments was correlated with changes in QoL. Instead, the aim of the present study is different and is to investigate near-death QoL (i.e., in the last two months before death) in advanced cancer patients receiving HPN. In addition, this secondary analysis is also based on new data from the analysis of questionnaires collected in the patients’ follow-up after the end of the study period of the previous study and until their death.

The goal of care in advanced cancer patients is usually not a cure, but symptom control, maintenance of acceptable QoL, and prolongation of survival [24]. QoL has been suggested as an indicator to assess the effectiveness of nutritional interventions in palliative care patients [7]. An expert panel determined QoL as the strongest outcome predictor for HPN in patients with cancer [25].

Many studies have reported a significant improvement in QoL in cancer patients receiving HPN. In 1993, King et al. first showed improvement in overall QoL, gastro-intestinal symptoms, fatigue, morale, and social interactions in 61 aphagic patients with gynecologic malignancy after 30 days of HPN [26]. Using KPS as an indirect indicator of QoL, Cozzaglio et al. found that QoL improved in 68% of patients treated for more than three months [27]. In a prospective observational study, Bozzetti et al. found QoL indexes stable up to two months prior to death in incurable aphagic patients undergoing total HPN [19]. Culine et al. observed significant improvement in QoL after 28 days from HPN start in 437 cancer patients [28]. Similarly, Vashi et al. reported a significant increase in QoL in 52 patients with cancer evaluated using the EORTC QLQ-C30 questionnaire after one, two, and three months [29]. Girke et al. observed advanced cancer patients on HPN for four weeks, who showed improvement in emotional and social functioning, dyspnea, sleeping, and QoL [30].

In our longitudinal study carried out in 111 cancer patients on HPN, several scores of the EORTC QLQ-C30 questionnaire (i.e., global QoL, role functioning, physical functioning, emotional functioning, fatigue, and loss of appetite) had a statistically significant improvement over the four months of evaluation [17]. Obling et al. in an RCT including incurable patients with gastrointestinal cancer showed that at 12 weeks, QoL was significantly improved in the HPN group compared with the control treatment [31].

Conversely, Bouleuc et al. in an RCT with 148 cancer patients on HPN did not observe significant changes in health-related QoL [32]. However, this study had some limitations that deserve consideration, the main one being that in the HPN arm, 46% of patients had an Eastern Cooperative Oncology Group (ECOG) PS 3 or 4 [33]. Thus, the inclusion criteria were not in accordance with the recommendations for HPN in the guidelines [18], indicating that HPN candidates should have a KPS > 50 (i.e., ECOG < 3).

Data from the literature showed that clinical characteristics at the HPN start were associated with improved QoL and survival in advanced cancer patients. An analysis of 969 palliative care patients with cancer on artificial nutrition at home (HAN), including 629 on HPN, showed that HAN was effective in improving or maintaining the KPS in 90% of cases at one month. In addition, the selection criteria made it possible to identify patients who can benefit from HAN on survival and PS [34]. In a large prospective cohort study we showed that there were predictive factors (e.g., KPS > 50) significantly correlated with survival of patients with cancer undergoing HPN [35]. Specific predictive models have been proposed to predict survival in incurable patients with cancer receiving HPN; however, the clinical insight of experienced clinicians is needed to complement the accuracy even of a validated model [36,37,38].

Physicians should select patients with cancer who could benefit from HPN and balance the expected benefits with HPN without extending survival in those who have no chance of improving [39]. Regarding the ethical perspective of this decision-making, there has been a great deal of discussion about whether palliative patients with cancer should be fed. This ethical dilemma constitutes a matter of controversy. In fact, although there are limited benefits, delivery of HAN to patients with cancer who are in the last weeks of life is a frequent occurrence [40].

Reduced food intake, appetite loss, weight loss, and especially cachexia are associated with a high prevalence of psychological symptoms, such as anxiety and depression, resulting in marked impairment of QoL and reduced performance status [41]. Discharge from the hospital and the burden of HPN can be a time of anxiety and depression for patients. However, these symptoms decreased as well as improvement in QoL, mental and physical health scores after one month and three months of HPN was observed [42]. In women with ovarian cancer, HPN was perceived as enabling them to prolong survival and improve QoL. Patients and their caregivers reported that the impact on activities of daily living caused by HPN was worth the extended time they spent at home [43].

Anxiety surrounding feeding and associated psychological distress have a negative influence on the patients’ and their loved ones’ QoL [44]. Nevertheless, attempting to get the patient to eat when they have a significant loss of appetite can result in increased patient distress in interactions with their family members/caregivers. Orrevall et al. conducted interviews with cancer patients on HPN and their family members [45]. This study showed that both respondents experienced psychological, physical, and social benefits from HPN, and these positive effects were generally found to outweigh the negative features linked to limitations in family and social interactions. Additionally, the authors reported the positive relationship between quality of care and QoL; in fact, patients who experienced an environment around them that was responsive to their needs perceived an improvement in their global well-being [45].

Health-related QoL is a multidimensional concept which quantifies the physical, psychological, and social aspects of an illness and its treatment [46]. In patients with cancer, health status is well reflected on the measured QoL, which is largely influenced by nutrition-related aspects [47]. QoL has been assessed with many patient-specific cancers questionnaires, some of which have been extensively validated, such as the EORTC QLQ-C30 used in our study [19].

The mechanisms through which HPN improved near-death QoL in our patients are not clearly identifiable. We often observe that patients and their family members perceive the benefits of HPN at the end of life. In some patients, HPN alleviates symptoms like nausea, fatigue, loss of appetite, and pain [17,29]. It is probable that the combined medical nutrition therapy and skilled home care of these patients may have played a significant role in the improvement of global QoL [25,47]. It is most reasonable to assume that this improved QoL is not mainly due to a single intervention (e.g., HPN), but to the patient’s coping with the illness itself and the patients’ and family members’ sense of relief and security resulting from meeting the patient’s nutritional needs., which overrides the emotional distress of “starving to death” [45,48].

### 4.1. Clinical Implications and Future Research

European guidelines recommend delivering medical nutrition therapies in advanced stage cancer patients only after considering several aspects, including the expected benefit on QoL [14]. The findings of this study could support this recommendation of the practical guideline.

Suggestions for future research directions could be many, e.g., testing interventions that combine HPN with other palliative care measures to optimize QoL; analyzing the ethical dilemmas surrounding the initiation or withdrawal of HPN in the near-death phase; and studying healthcare utilization patterns, including hospitalizations, among HPN patients.

### 4.2. Strengths and Limitations

The novelty of the present study is to investigate such a controversial topic as near-death QoL in incurable cancer patients on HPN. It appears to address a gap in the knowledge body on QoL in these patients, specifically whether HPN is useful in improving QoL even in the last two months before death. The point of strength is the careful management of patient care, which made it possible to follow all of them until death and to maintain good quality home care.

This study has several limitations, the most important of which is that we performed a secondary analysis of previously collected data. However, the secondary analysis of quantitative data is a common and generally accepted mode of inquiry. In particular, the approach may be employed by researchers to reuse their data and generate new knowledge in the case of data from inaccessible respondents [49]. A second limitation is the absence of a control group. Nevertheless, an RCT is ethically not acceptable since a control group of severely hypophagic patients not receiving any medical nutritional therapy are at risk of an earlier death from malnutrition rather than cancer progression. In addition, the sample size of this observational trial was not calculated a priori, and the number of its participants was only based on the fixed length of the study (two years). Finally, this study describes the experience at large, tertiary care hospitals with well-developed, multidisciplinary hospital and home care teams, which may limit generalizability of findings, particularly to settings with different healthcare resources.

## 5. Conclusions

Cancer often leads to progressive deterioration of QoL in advanced patients. Providing HAN to patients with cancer who are in the last months of their life is a controversial issue. This study first reports that HPN, if started early and according to European guidelines, is associated with an improvement in the QoL of patients with cancer even in the last two months of life.

The results of our study may have practical implications for clinicians managing cancer patients on HPN. In particular, the results may support physicians in their decision to prescribe HPN even to patients with incurable cancer with a life expectancy of few months, because HPN can improve their near-death QoL.

## Figures and Tables

**Figure 1 nutrients-17-00271-f001:**
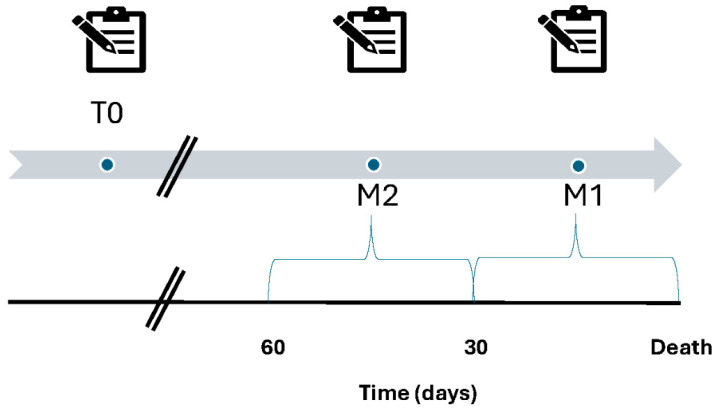
Timeline of questionnaire completion. T0: the questionnaires filled out at home at parenteral nutrition start. M2: the questionnaires filled out between 31 and 60 days before death. M1: the questionnaires filled out in the 30 days before death.

**Table 1 nutrients-17-00271-t001:** Patients’ characteristics.

*N*	84
Female gender, *n* (%)	40 (47.6)
Age (years), median (range)	62 (32–81)
Tumor site, *n* (%)	
Stomach	29 (34.5)
Pancreas/biliary system	17 (20.3)
Colon/rectum	17 (20.3)
Esophagus	7 (8.3)
Lung	6 (7.1)
Others	8 (9.5)
Stage, *n* (%)	
III	19 (22.6)
IV	65 (77.4)
Metastasis, *n* (%)	58 (69.0)
Karnofsky PS, median (range)	70 (70–80)
BMI, median (range)	20.5 (13.5–29.5)
Weight loss ^1^ (%), median (range)	12.2 (2.1–38.3)
PG-SGA category, *n* (%)	
A	1 (1.2)
B	29 (34.5)
C	54 (64.3)
Oral food intake (Kcal), median (range)	600 (250–1350)
Indication for HPN, *n* (%)	
Intestinal (sub)obstruction ^2^	66 (78.6)
SBS; high-output ileostomy or fistula	11 (13.1)
EN not tolerated or feasible	7 (8.3)

PS, performance status; BMI, body mass index; PG-SGA, patient-generated subjective global assessment; SBS, short bowel syndrome; EN, enteral nutrition; HPN, home parenteral nutrition. ^1^ In the last 3 months before HPN. ^2^ Intra-abdominal recurrence and/or peritoneal carcinomatosis.

**Table 2 nutrients-17-00271-t002:** Patients’ scores for the EORTC QLQ-C30 scales at different time-points.

Item	T0	M2	*p* M2 vs T0	M1	*p* M1 vs T0
EF ^1^	50.0 (39.1–68.8)	46.9 (37.5–50.0)	0.555	43.8 (37.5–56.3)	0.115
RF ^1^	75.0 (50.0–87.5)	56.3 (37.5–87.5)	0.328	62.5 (40.9–75.0)	0.520
CF ^1^	37.5 (25.0–50.0)	37.5 (37.5–62.5)	0.933	50.0 (25.0–60.0)	0.803
SF ^1^	37.5 (25.0–62.5)	37.5 (37.5–62.5)	0.783	37.5 (28.8–62.5)	0.970
PF ^1^	70.0 (50.0–87.5)	55.0 (45.0–76.3)	0.079	60.0 (50.0–80.0)	0.277
GQoL ^1^	42.9 (28.6–57.1)	71.4 (57.1–76.8)	0.001	64.3 (57.1–71.4)	0.035
FI ^2^	25.0 (25.0–50.0)	25.0 (25.0–50.0)	0.987	25.0 (25.0–50.0)	0.818
FA ^2^	83.3 (68.8–91.7)	75.0 (75.0–83.3)	0.148	83.3 (75.0–91.7)	0.684
NV ^2^	50.0 (37.5–62.5)	50.0 (37.5–62.5)	0.550	50.0 (48.1–75.0)	0.893
AP ^2^	75.0 (75.0–100.0)	75.0 (50.0–100.0)	0.084	75.0 (75.0–100.0)	0.111
SYMP ^2^	51.3 (45.4–60.7)	45.5 (38.5–55.4)	0.033	48.7 (43.9–57.7)	0.055

T0: questionnaires completed at start of home parenteral nutrition; M2 questionnaires completed between 31 and 60 days before death; M1 questionnaires completed between 0 and 30 days before death. ^1^ Functioning scales: scores range from 0 to 100, with a higher score representing a higher level of functioning. ^2^ Symptom scales: scores ranging from 0 to 100, with a higher score representing a higher level of symptoms. Scores are indicated as median value (25th–75th percentile). Statistical analysis was performed using the Mann Whitney’s U test. EORTC QLQ-C30: European Organisation for Research and Treatment of Cancer Quality of Life Questionnaire Core 30; EF, emotional functioning; RF, role functioning; CF, cognitive functioning; SF, social functioning; PF, physical functioning; GQoL, global quality of life; FI, financial impact; FA, fatigue; NV, nausea and vomiting; AP, appetite loss; SYMP, symptoms scale.

## Data Availability

The data that support the findings of this study are available from the corresponding author on reasonable request.

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
