# Peer review of "Near-Death Quality of Life in Cancer Patients on Home Parenteral Nutrition"

_nutrients, 2025, doi:10.3390/nu17020271_

Round 1

Reviewer 1 Report (New Reviewer)

Comments and Suggestions for Authors

The authors present some interesting data on patients who received HPN around the time of their demise, and who had filled out QOL forms.

This is an interesting report and may help understand the role of PN in end of life care fro some patients. I have a few comments:

1. The tittle is a bit confusing. Truly is looking at QOL in patient near end of life or near death. but is not near death QUOL as the last report is within a month an a month is a long time in this situations

2. A figure explaining the sequence is the data collection maybe helpful. one would expect to go from T0 to M1 to M2, but actually is T0 to M2 to M1. and reading it is confusing

3. Time of HPN supplementation needs to be reported as one could not expect same changes in QOL if patient receives HPN for shorter time than for longer periods.

4. As this is a report on a prospective series without a control group, how can the authors be sure that HPN over IV Fluids alone, or  over home health care  care alone are not related with improved QOL, as the palliative care data has shown over and over that palliative services improve QOL at end of life.

5. Where any deaths related with HPN? Sepsis? major electrolyte disorders i.e hyperkalemia.  

Author Response

The authors present some interesting data on patients who received HPN around the time of their demise, and who had filled out QOL forms.

This is an interesting report and may help understand the role of PN in end of life care fro some patients. I have a few comments:

  1. The tittle is a bit confusing. Truly is looking at QOL in patient near end of life or near death. but is not near death QUOL as the last report is within a month an a month is a long time in this situations.

R. Thank you for pointing out this important point. I completely agree with your comment that “a month is a long time”. However, the ESPEN guidelines ‘Home Parenteral Nutrition’ state that “HPN should be prescribed to prevent an earlier death from malnutrition in advanced cancer patients with CIF, if their life expectancy related to the cancer is expected to be longer than one to three months, even in those not undergoing active oncological treatment”.

For this reason, as the guidelines explicitly refer to the death of the patient with advanced cancer, the aim of the present study is to explore whether HPN is useful for improving QoL even during the last two months before death. This is precisely why we wanted to draw the reader's attention in the title to this particular phase in the life of the patient with advanced cancer (near-death) and the HPN-related QoL.

  1. A figure explaining the sequence is the data collection maybe helpful. one would expect to go from T0 to M1 to M2, but actually is T0 to M2 to M1. and reading it is confusing

R. Thank you for the suggestion. We have added Figure 1 explaining the timeline of questionnaire completion.

  1. Time of HPN supplementation needs to be reported as one could not expect same changes in QOL if patient receives HPN for shorter time than for longer periods.

R. Thank you for the suggestion. We have added the median duration of HPN at the end of the paragraph ‘Patients’ in the ‘Results’.

  1. As this is a report on a prospective series without a control group, how can the authors be sure that HPN over IV Fluids alone, or over home health care care alone are not related with improved QOL, as the palliative care data has shown over and over that palliative services improve QOL at end of life.

R. Thank you for pointing out this important point. A limitation of the present study is the lack of a control group. However, an RCT is ethically unacceptable because a control group of patients who are severely hypophagic and do not get any nutritional support is at risk of earlier death due to malnutrition rather than from cancer progression. We have added this limitation in the ‘Discussion’.

  1. Where any deaths related with HPN? Sepsis? major electrolyte disorders i.e hyperkalemia.

R. Thank you for the suggestion. We have added these data at the end of the paragraph ‘Patients’ in the ‘Results’.

Reviewer 2 Report (New Reviewer)

Comments and Suggestions for Authors

Thank you for the opportunity to review this important study.

I kindly ask for a verification of the data's consistency between the chapter: 2.2. Procedures  “a need of 20–25 kcal/kg/day bedridden or 25–30 kcal/kg/day for ambulatory patients and 1–1.5 g/ kg/day aminoacids.” 

and the data presented in the chapter : 3.1. Patients “therefore, supplemental HPN provided a median amount of 1000–1250 kcal per day.”.

 Conclusion chapter:
This sentence : “Further studies are needed to better understand this much-debated topic of end-of-life care” 

seems obvious and unnecessary. I kindly request you to consider removing it from the study text.

Author Response

Thank you for the opportunity to review this important study.

I kindly ask for a verification of the data's consistency between the chapter: 2.2. Procedures  “a need of 20–25 kcal/kg/day bedridden or 25–30 kcal/kg/day for ambulatory patients and 1–1.5 g/ kg/day aminoacids.” and the data presented in the chapter : 3.1. Patients “therefore, supplemental HPN provided a median amount of 1000–1250 kcal per day.”.

R. Thank you for pointing out this important point. The caloric requirements of the patients enrolled in this study were between 1600 and 1800 kcal per day. But because all patients had a median residual oral food intake of 600 kcal, the supplemental HPN program provided a median amount of 1000-1250 kcal per day.

Conclusion chapter:

This sentence : “Further studies are needed to better understand this much-debated topic of end-of-life care” seems obvious and unnecessary. I kindly request you to consider removing it from the study text.

R. Thank you for the suggestion. We have removed the sentence.

This manuscript is a resubmission of an earlier submission. The following is a list of the peer review reports and author responses from that submission.

Round 1

Reviewer 1 Report

Comments and Suggestions for Authors

Congratulations to the authors for your work assessing the quality of life of patients with HPN. As mentioned in the manuscript, this is a continuation of a previously published article. This point raises several questions about the present manuscript. 

The authors have tried to make a second analysis from a prior published work. Doing a second post hoc analysis is not easy and, sometimes, it is hard to demonstrate how proper is the second analysis.

Replicating or analyzing post hoc the prior study could be fine as long as the authors make it clear where the original idea or theory comes from. Since the manuscript heavily bases the present manuscript on previous research, the authors should mention this in the abstract, as well as in the introduction and methods. Despite the prior study has been mentioned, the reasons why the results in the manuscript deserve to be published or amplified and were not published or included in the prior study have not been stated.

From my point of view, this manuscript might have low research value. The main reason is the fact that the data come from a prior publication and no new data are shown.  

Short comments related to the manuscript:

Data from patients who died included in the prior publication were included  (84 of 111 patients; 75.7%). In addition, the data analyzed have been the same as the prior publication, comparing the first and the last results from the questionnaire. Therefore, no additional data have been given in the present manuscript.  Moreover, according to the prior study, " died during the study period of QoL analysis, 24 of 72 (33%) were on oncologic treatments and 23 of 39 (59%) were without oncologic treatments."; however, the present manuscript declared that 84 who died were included. This point should be clarified by the authors giving reasons for this difference. 

The authors stated only non-parametric methods; why? did the authors expect these results? A rationale for this statement should be added.

Table 2 is shown as 2 blocks, and T0 column is repeated. This table should be unified and T0 data should be shown only once. 

Given that no new data from the study is given, I think there is no reason to duplicate the publication.  There is indeed a new analysis of a subgroup of patients. However, the present manuscript is presented in such a small analysis that it might be not enough for being a "Nutrients" paper. Thus,  I suggest it could be fine for a short communication in another journal. 

To sum up, I find that few data are declared in the manuscript to deserve to be published in the present format. I recommend the authors review their data and add new ones if they want to rebuild a new manuscript related to the prior study.  

Reviewer 2 Report

Comments and Suggestions for Authors

The current manuscript describes a retrospective cohort study that looked for the changes in quality of life during the last two months before death compared to the quality of life at the start of a home parenteral nutrition (HPN) regimen, among cancer patients. This study is novel and important from the perspective of improving the nutrition-associated quality of life during end-of-life and palliative care regimens. The title and the abstract adequately reflect the study content!

The introduction section appears well-written. This section gradually introduces the topic, describes the problem statement and current state of affairs, and eventually flows to the inquiry question mentioned at the end of the section. This section does a good job of expanding on the complex relationships between malnutrition and cancer, as well as how they reinforce one another and spiral toward diminished quality of life. The introduction section discusses the relationship between both conditions and adequately discusses the intricacies of the topic. The inquiry section adequately describes the purpose and the objectives of the current study in terms of whether HPN is really helpful in improving the quality of life during the last two months before death.

The methods section adequately describes the study plan. This study used data collected from two tertiary referral care, university-affiliated hospitals. Trained professionals and statisticians have conducted the analysis. All questionnaires used in the study have been adequately described! The statistical analysis section appears to be robustly done using R software version 3.5.3.

The result section describes the study's findings and concurs with the information presented in the tables. The tables appear clear and legible!

The discussion section adequately summarizes the main findings of the study. It has summarized the findings and elaborated on them and compared and contrasted them with findings in similar studies or research settings.

The conclusion section adequately summarizes the study findings and discusses their implications for changes in practice!

In general, this study is well-written and ready for publication in its current form!

Reviewer 3 Report

Comments and Suggestions for Authors

As a peer review editor for a Gastroenterology journal, I have carefully reviewed the manuscript titled "Near-death quality of life in cancer patients on home parenteral nutrition." Below are my detailed comments and suggestions for revision:

Title

The title is clear and relevant to the study's content. It accurately reflects the focus on near-death quality of life (QoL) in cancer patients receiving home parenteral nutrition (HPN).

Abstract

The abstract provides a concise summary of the study.

Introduction

The introduction is well-written and provides a comprehensive background on malnutrition in cancer patients and the role of HPN. Please,  Clearly identify the gap in the existing literature regarding near-death QoL and the rationale for this study.

Methods

The methods section is detailed and comprehensive. However, a few areas need clarification:

Study Design: Clearly state that this is a secondary analysis of a previous study. Mention the timeframe of the original study.

Sample Size and Selection: Provide a justification for the sample size and describe the inclusion and exclusion criteria in more detail.

HPN Protocol: Include more details about the HPN protocol, such as specific nutritional components and administration guidelines.

Ethical Considerations: Expand on the ethical considerations and consent process.

Results

The results section presents the findings clearly.

Discussion

The discussion is well-structured and places the findings in the context of existing literature. However, it could be strengthened by:

Interpretation of Results: Provide a more detailed interpretation of the findings, particularly the mechanisms through which HPN might improve QoL near death.

Clinical Implications: Elaborate on the clinical implications and how these findings could influence practice guidelines.

Future Research: Suggest areas for future research based on the findings and limitations.

Conclusion

The conclusion succinctly summarizes the main findings. However, Highlight the practical implications for clinicians managing cancer patients on HPN.

Additional Comments

  1. Language and Style: The manuscript is well-written, but a thorough proofreading for minor grammatical errors and consistency in terminology is recommended.
  2. Figures and Tables: Ensure all figures and tables are of high quality and add value to the manuscript.

Overall, this manuscript addresses an important and under-researched area in the field of oncology and nutrition. With revisions to clarify the methodology, enhance the discussion, and strengthen the overall narrative, this study has the potential to make a significant contribution to the literature on QoL in cancer patients receiving HPN.

Please revise the manuscript accordingly and resubmit for further review.

Comments on the Quality of English Language

This manuscript is well-written, but a thorough proofreading for minor grammatical errors and consistency in terminology is recommended.
